



# $X\mathrm{CO_2}$ observations compared to km-scale ICON-ART simulations indicate an underestimation of Thessaloniki's emissions in the ODIAC inventory

Lena Feld[1], Pablo Schmid[1], Marios Mermigkas[2,a], Dimitrios Balis[2], Jochen Gross[1], Darko Dubravica[1], Carlos Alberti[1], Benedikt Herkommer[1], Stefan Versick[1], Roland Ruhnke[1], Frank Hase[1], and Peter Braesicke[1,b]

[1]Institute of Meteorology and Climate Research Atmospheric Trace Gases and Remote Sensing, Karlsruhe Institute of Technology, Karlsruhe, Germany
[2]Laboratory of Atmospheric Physics, Department of Physics, Aristotle University of Thessaloniki, Thessaloniki, Greece
[a]now: Institute for Astronomy, Astrophysics, Space Applications and Remote Sensing, National Observatory of Athens, Athens, Greece
[b]now: Deutscher Wetterdienst, Offenbach, Germany

**Correspondence:** Lena Feld (lena.feld@kit.edu)

**Abstract.** An accurate inventory of $CO_2$ emissions is important for the implementation of effective reduction measures and thus for climate change mitigation. Most current inventories are based on reported activities and rely little or not at all on atmospheric data. However, these inventories have large uncertainties, especially for smaller scales such as urban areas. For example, for the city of Thessaloniki, Greece, the EDGAR inventory reports 3.1 Mt, which differs by 72 % from the emission
estimate of the ODIAC inventory (1.8 Mt) for the same area for the year 2019. With a measurement campaign in the framework of the Collaborative Carbon Column Observing Network (COCCON), we collected observations for three months in October 2021 and summer 2022 in Thessaloniki. A total of 30 days of column averaged molar fractions of $CO_2$ ($X\mathrm{CO_2}$) were recorded. We combine these data with km-scale simulations from the numerical weather prediction model ICON-ART. The ODIAC inventory was used for simulating the emission of $CO_2$. We optimized the simulated atmospheric time series of $X\mathrm{CO_2}$ to best
match the observed data by scaling the prior emissions using a least-squares approach. With different configurations, we found a consistent up-scaling of the prior emissions, with total emissions ranging from 2.9 to 4.4 Mt in the urban area of Thessaloniki. This estimate is significantly higher than the emissions reported in ODIAC. The result demonstrates the potential of including ground-based column measurements of $CO_2$ in the construction of emission estimates to reduce uncertainties at the urban scale.

## 1 Introduction

Rapidly reducing carbon dioxide ($CO_2$) emissions is key to mitigating climate change. Being able to monitor emissions and their changes over time is important for reliable stocktaking and taking targeted action. Currently, most emission inventories



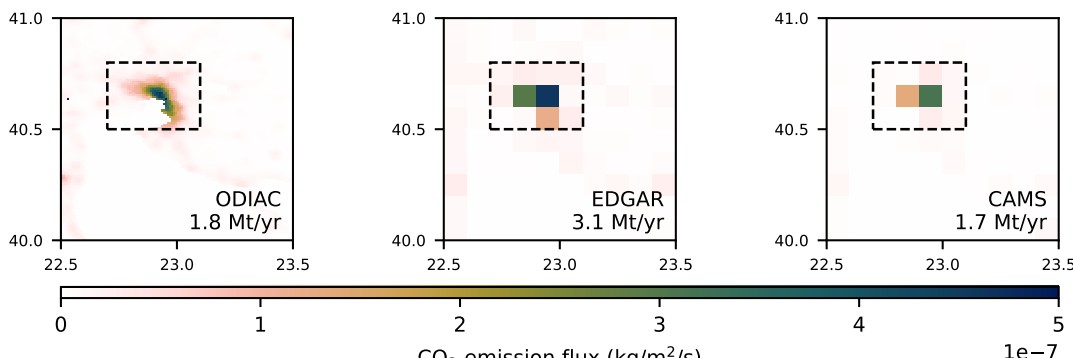

**Figure 1.** The average anthropogenic $CO_2$ emissions of Thessaloniki in 2019 from three inventories. Left: The spatially well-resolved ODIAC inventory that will be used in this paper. The EDGAR inventory (center) and the CAMS inventory (right) are spatially more aggregated than the ODIAC inventory. Note the differences in total emissions for the city area (defined by the dashed rectangle) displayed in the lower right corner of each frame.

are based on official reporting and show large differences, especially when looking at individual regions (Super et al., 2020; Solazzo et al., 2021).

Therefore, a combination of different measurement and modeling approaches is needed to reduce the uncertainties in our current emission estimates. In particular, urban areas are of importance due to their significant localized contributions to the global emissions. In 2015 a third of the global anthropogenic greenhouse gas emissions was produced in urban areas and this fraction has increased over the past decades (Crippa et al., 2021). At the same time, the complex heterogeneous structure of cities makes it difficult to attribute observed greenhouse gas concentration changes to emission sources. Here, we investigate

the emissions of the urban area of Thessaloniki using a small targeted measurement campaign with two portable Fourier Transform Infrared (FTIR) spectrometers and the state-of-the-art ICON-ART model in a local-area setup. With 1.09 million inhabitants, Thessaloniki is the second-largest city of Greece (Eurostat, 2023). Its current $CO_2$ emissions are estimated in different inventories.

Comparing existing inventories for the area of Thessaloniki reveals the considerable discrepancies. Three different emission

inventories – EDGAR, CAMS and ODIAC – are shown in Figure 1. The emissions of the EDGAR v7.0 inventory (Branco et al., 2022) are derived by summing up reported $CO_2$ emitting activity multiplied by emission factors for the respective processes. Solazzo et al. (2021) state an uncertainty of 7% for the global $CO_2$ emissions. However, when looking at individual grid cells of the inventories, the uncertainties become much larger. Super et al. (2020) find an uncertainty of up to 40% when considering a central European domain. The Copernicus Atmosphere Monitoring Service (CAMS) is publishing a separate

emission inventory that is constructed by combining the EDGAR v4.3.2 and CEDS datasets (Granier et al., 2019). Both EDGAR and CAMS inventories have a horizontal spatial resolution of 0.1 x 0.1 degree. The spatial heterogeneity and shape of cities is not well resolved at such a resolution. This also poses a challenge for the correct attribution of localized sources. Here, cities map into a single or a few pixels. For the high-resolution ODIAC inventory with a resolution of 1 km (EDGAR





and CAMS are in the order of 10 km), data from the Carbon Dioxide Information Analysis Center (CDIAC) was spatially
disaggregated using satellite data of nighttime artificial light sources, geographical location of point sources and aircraft and
ship fleet tracks (Oda and Maksyutov, 2015). To compare the total emissions of the urban area of Thessaloniki, we defined a
*city area* ranging from 40.5 to 40.8 °N and from 22.7 to 23.1 °E. The city area is indicated by the dashed rectangle in Figure 1,
with the total emissions of this area are displayed in the corresponding panels. The comparison between the three inventories
clearly shows that beyond the different resolutions also the spatial distributions and the total emissions deviate. EDGAR reports
3.1 Mt in 2019 for this area, which is about double the emissions reported by CAMS (1.7 Mt/yr). It also differs by 72 % from
the ODIAC inventory, which reports 1.8 Mt per year. This supports our rationale that novel measurement-based methods and
state-of-the-art models are needed to verify and improve the reporting-based emission estimates.

One suitable measurement technique for the quantification of greenhouse gas amounts on regional scales is solar FTIR spec-
troscopy using the portable EM27/SUN spectrometer developed by Bruker and KIT (Gisi et al., 2011; Hase et al., 2016). In the
scope of the Collaborative Carbon Column Observing Network (COCCON) – a framework for EM27/SUN operation (Alberti
et al., 2022; Frey et al., 2019) – several urban measurement campaigns have been performed in the past using multiple instru-
ments to derive emissions from gradients between instruments. The COCCON is enabled by ESA and defines instrumental and
data analysis standards.

Generally, to estimate emissions from campaign observations, a model that connects the target quantity (emission flux) with
the observed quantity (column-averaged molar fractions) is required. A variety of different approaches has been used for this
purpose in the past. Simplified models, for example a box-model (Chen et al., 2016; Makarova et al., 2021) or plume model (Tu
et al., 2022) were used. Hase et al. (2015) numerically implemented a Lagrangian dispersion model, dividing the city of Berlin
into different rectangles with separate source strength, using a simplified wind field for the city. More complex meteorological
transport models were used for the campaigns in Paris and Tokyo. These models numerically predict the $X\mathrm{CO_2}$ abundances at
the observing sites using a prior emission map. For the interpretation of the data recorded in Paris by Vogel et al. (2019), $X\mathrm{CO_2}$
time series were modeled using the chemistry-transport model CHIMERE. The comparison between simulated and observed
data was discussed qualitatively. For the Tokyo campaign (Ohyama et al., 2023), trajectories were modeled with HYSPLIT.
Emission estimates were derived from the observed data using Bayesian inversion. A complementary approach was used for
a campaign investigating dairies in the Los Angeles Basin. Instead of calculating trajectories, Viatte et al. (2017) use the
numerical weather prediction (NWP) model WRF in a large eddy simulation (LES) configuration for simulating the emissions
and transport of Methane. They also derive posterior emission estimates by using a Bayesian inversion approach.

## 2   Data and methods

Here, the results of a campaign with 221 total hours of measurement time (excluding calibration) in the urban area of Thes-
saloniki, Greece, will be analyzed. The complex orography and the land-sea wind system considerably complicate the task of
deriving emissions and require the analysis to include accurate weather and transport simulations. We will utilize a local-area
setup of the NWP model ICON-ART (Zängl et al., 2015; Schröter et al., 2018) to create regional hindcasts using emissions





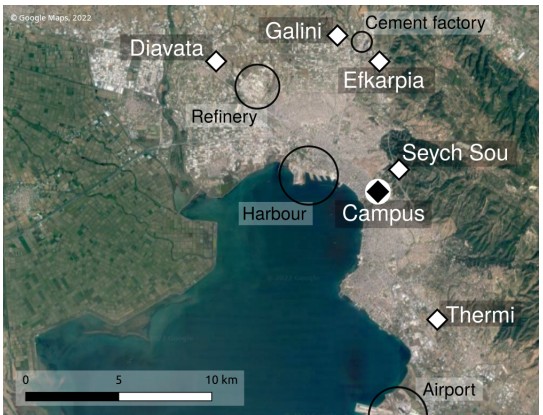

**Figure 2.** Observation sites for the Thessaloniki campaign. All measurement sites are marked with diamonds on the satellite image. The site for the continued long-term measurements on the AUTh campus in the central part of the city is marked in black while the sites that were used for the second spectrometer are marked in white. In addition, some expected significant source regions are indicated by black circles. The satellite image is taken from © Google Maps (2022).

from the high resolution ODIAC inventory. Instead of a Bayesian inversion approach, we use a decomposition of the emissions. Least-squares optimization is used to re-weight the emissions of individual pixels in the urban area to derive emission estimates from the observational data.

## 2.1 Observational dataset

**Data collection**

The EM27/SUN is a solar absorption FTIR spectrometer. Among other species, it records the column-averaged dry-air mole fraction of carbon dioxide, denoted $X\text{CO}_2$. The spectrometer can deliver data with a temporal resolution of up to a few seconds. Here, apply a temporal resolution of 1 minute. The average standard deviation of $X\text{CO}_2$ is 0.13 ppmv for spectrometers of the
considered ensemble (Frey et al., 2019).

$X\text{CO}_2$ is derived from the vertical columns ($VC$) of carbon dioxide and oxygen.

$$X\text{CO}_2 \quad = \quad \frac{VC_{\text{CO}_2}}{VC_{\text{dry air}}} \tag{1}$$

$$= \quad \frac{VC_{\text{CO}_2}}{VC_{\text{O}_2}} \cdot f_{\text{O}_2} \tag{2}$$

where the ratio of oxygen is $f_{\text{O}_2} = 0.2095$. In a first step, slant columns are retrieved from the recorded solar spectra. They are
corrected for the solar zenith angle of the observation and transformed into vertical columns. Additional air mass dependent and air mass independent calibration factors as well as an empirical $\text{H}_2\text{O}$ correction are applied (see Herkommer (2024) for more details). These corrections on the sub-percent level compensate for residual spectroscopic uncertainties.



**Table 1.** Available datasets of meteorological measurements. Measurement site A is only recording pressure. For the others, also wind information is available. All measurements are recorded with an interval of 10 min, except D which is only available at hourly intervals.

|   |   | Site | Accuracy of p in hPa |
|---|---|---|---|
| A | Vaisala (PTB330) | Campus (Physics) | 0.15 |
| B | Davis (Vantage Pro2) | Thermi | 1.0 |
| C[1] |   | Campus (Meteor.) |   |
| D[2] | WMO (Station 16622) | Airport |   |

[1] https://meteo.geo.auth.gr/en/data-availablity/

[2] https://meteostat.net/de/station/16622

Similar to previous COCCON campaigns, spatial gradients of $X\text{CO}_2$ were observed by positioning multiple spectrometers at different sites in or around the city. For the campaign in Thessaloniki, two spectrometers were used. The Thessaloniki long term measurement by Mermigkas et al. (2021) with the spectrometer *SN52* was continued at the Campus located in the city center. The spectrometer is placed on the roof of the school of physics building, on campus of the Aristotle University of Thessaloniki. The campus is located in the center of Thessaloniki, in a highly polluted part of the city, with a severe traffic load. For mobile use, we equipped the spectrometer *SN96* with a solar power supply, using two portable solar panels and a battery for buffering. This allows us to cover a large area of the city, by measuring with this portable spectrometer at different positions on a daily basis. All measurement sites are shown in Figure 2. The data processing was done with the retrieval algorithm PROFFAST 2.3 using the PROFFASTpylot interface (Hase, 2022; Feld et al., 2024). The two spectrometers were calibrated by side-by-side measurements performed at the beginning, middle and end of the campaign period (see Section 3.1).

**Additional data sources**

Auxiliary meteorological observations are needed for the data processing and interpretation. For the retrieval of $X\text{CO}_2$ from the spectra, precise pressure records at the location of the spectrometer as well as prior information of the atmosphere is needed. The available pressure data sources are summarized in Table 1. Due to significant time gaps in the pressure recording A, we constructed a continuous pressure dataset by combining measurements A and B. This combined pressure is labeled as *reference pressure* in the following. Since the pressure is needed at the height of the observation, we used a portable pressure sensor that we transported between the sites to derive pressure factors converting the reference pressure to the pressure at observation height. The datasets B, C and D also contain wind information. They are compared to the simulated wind for validation of the forward simulations with ICON-ART.





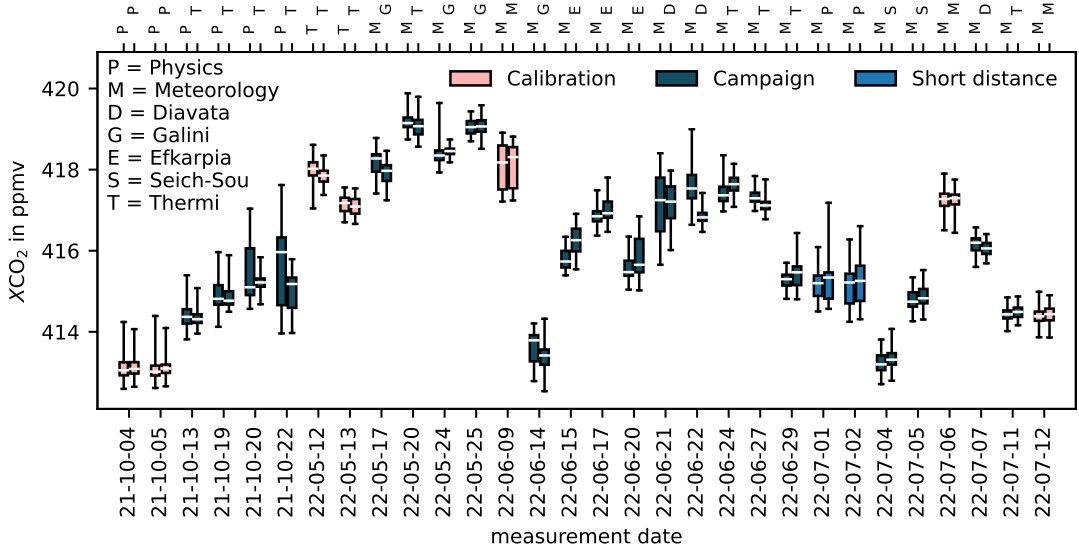

**Figure 3.** Overview of the observational dataset. $XCO_2$ is shown for each day from both spectrometers as box-whisker plot. The box extends from the 25 to the 75 percentile, the whiskers mark the minimum and maximum observed value at the given date and the median is marked by a white line. The locations are color-coded (see legend on the upper left) and in addition, the first letter of the locations are attached to the top axis.

### Dataset overview

In total 30 days of measurements were recorded. From these, 7 days were used for side-by-side measurements to calibrate the two spectrometers in regular intervals. The uncertainty of the scaling factors for calibrating the two spectrometers is among the fundamental error sources in this study. We discuss the relative calibration uncertainty in Section 3.1. The remaining 23 days were recorded with the mobile unit being operated at different locations in the city. One of the days was excluded, because no initial meteorological data was available for this day to perform an corresponding simulation (see Section 2.2). The 22 days selected for further analysis contain 179 hours of measurements, when taking both spectrometers into account. Two of the days were dedicated to measurements with only a small distance between the stationary and the movable EM27/SUN to test the small-scale variability (see Section 3.1). An overview over all 30 days is shown in Figure 3.

The daily variations (i.e. the maximum observed difference in $XCO_2$ by a spectrometer at a given day) ranged from 0.56 ppmv to 3.66 ppmv, with a median daily variation of 1.33 ppmv. When looking at the gradient between both spectrometers $\Delta XCO_2$, the maximum observed difference is 2.03 ppmv while the median of $|\Delta XCO_2|$ amounts to only 0.17 ppmv. In conclusion, mainly small variations below the 1 ppmv level are present in the observed dataset, while previous campaigns targeting larger cities recorded $XCO_2$ signals up to several ppmv. For example Ohyama et al. (2023) observe gradients up to 9.5 ppmv. Still, the variations observed in Thessaloniki are significant and carry information about the local sinks and sources (compare Section 3.1).



## 2.2 Simulation setup

As explained above, the daily variation measured in the dataset are relatively small compared to other campaigns. Wind speeds
and directions are changing too much over the day and between different locations to interpret the variations using stronger
simplified model approaches as the box model (as shown by Schmid (2023)). A precise simulation of the local transport and
emission characteristics of $CO_2$ is therefore mandatory to interpret the the observations. For this purpose the numerical weather
forecast model ICON-ART (version 2.6.5) developed by the German weather service (DWD), MPI-M and KIT (see Zängl et al.
(2015); Schröter et al. (2018)) is used.

### Domain and resolution

The setup is similar to the one used for the operational forecast over Germany with a 2 km resolution. However, a denser
simulation grid with a resolution of 1 km was configured for this setup, which is approximately the resolution of the ODIAC
inventory that was used as the emission source to be tested. More precisely, if defined by the square root of the cell area, the
R19B7 grid used for the ICON simulations has a resolution of 1.033 km (see Zängl et al. (2015) for the grid nomenclature).
The average resolution of the ODIAC inventory at the latitude of Thessaloniki is 0,808 km.

The simulations are performed in a local-area setup with a regional grid centered on Thessaloniki (40.6 °N, 23.0 °E) with a
width of 2 degree in longitudinal direction and 1.6 degree in latitudinal direction.

### Initialization

Initial and boundary data were derived from the operational European forecast of the DWD at a resolution of 6.5 km. Assuming
that most of the daily variability observed in the $CO_2$ time series is due to local emissions, we set the surrounding background
concentration to 0 ppmv, so that only the enhancement of the $CO_2$ mixing ratio from sources within the domain is simulated. To
be able to neglect the background concentration, the tracers are simulated as passive tracers, so that no concentration-dependent
depletion is taken into account. For each measurement day, a separate simulation was run, starting at 3 UTC.

### Sources and sinks

Here, we use the ODIAC inventory for the anthropogenic $CO_2$ emissions, because of its high spatial resolution. Although the
spatial resolution of the simulation grid and the ODIAC grid are both approximately 1 km, a re-mapping is needed. The rectan-
gular ODIAC grid structure is re-mapped to the triangular grid structure of the ICON simulation grid using emiproc (C2SM-
RCM, 2023).

For the biogenic contribution, an estimate for the net ecosystem exchange (NEE) was constructed from three data sources.
The spatial distribution was taken from the *MODIS* NEE dataset, that has a lateral resolution of 1 km and a temporal resolution
of one year (Running and Zhao, 2021). The MODIS NEE fluxes are shown for the area around Thessaloniki in Figure 4A.
To obtain a better temporal resolution, the MODIS dataset was scaled to match the daily emissions reported in the *SMAP*





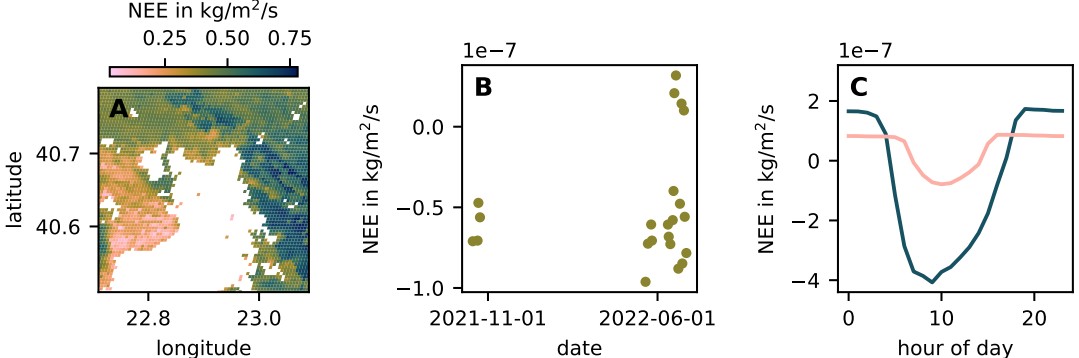

**Figure 4.** The datasets combined to simulate the biogenic NEE to the net emissions. The spatial distribution of the MODIS dataset is shown in panel A, the daily mean values from the SMAP dataset in panel B and the diurnal cycle for October 2021 and May 2022 from the FLUXCOM X-BASE dataset in panel C.

dataset by Kimball et al. (2022) (Figure 4B) and multiplied by an average diurnal cycle derived for the simulated area from the *FLUXCOM X-BASE* dataset provided by Weber (2023) (Figure 4C).

**Derived observation quantities**

The observed column-averaged dry-air mole fractions $X\mathrm{CO_2}$ is not directly calculated by ICON-ART and needs to be derived from quantities represented in the model. It can be calculated from the simulated wet-air volume mixing ratio $X_{\mathrm{CO_2}}^{\mathrm{grid\ cell}}$ at the location of the grid cell, by integrating over the model levels to obtain the total column of $CO_2$. Following Equation 1, $X\mathrm{CO_2}$ is derived from the vertical columns of $CO_2$ and dry air. The vertical column of a species A is defined by the number density $\rho_{\mathrm{A}}^{\mathrm{N}}$


$$VC_{\mathrm{A}} = \int_0^\infty \rho_{\mathrm{A}}^N \, \mathrm{d}z \quad . \tag{3}$$

For $CO_2$, this can be expressed from the pressure $p$ and temperature $T$ which are available from the model output.

$$VC_{\mathrm{CO_2}} = \frac{N_A}{R} \int_0^\infty \frac{p}{T} X_{\mathrm{CO_2}}^{\mathrm{grid\ cell}} \, \mathrm{d}z \quad . \tag{4}$$

The vertical column of dry air can be derived from the surface pressure $p_{sfc}$ and the vertical column of water vapor $VC_{H_2O}$

$$VC_{\mathrm{dry\ air}} = \frac{p_{sfc} \cdot N_A}{g \cdot M_{\mathrm{dry\ air}}} - VC_{H_2O} \frac{M_{H2O}}{M_{\mathrm{dry\ air}}} \quad , \tag{5}$$

using the molar masses of dry air $M_{\mathrm{dry\ air}} = 28.9 \, \frac{\mathrm{g}}{\mathrm{mol}}$ and of water vapor $M_{\mathrm{H_2O}} = 18.0 \, \frac{\mathrm{g}}{\mathrm{mol}}$. The vertical column of water vapor can be derived from Equation 3 as

$$VC_{\mathrm{H_2O}} = N_A \int_0^\infty \frac{q_v \rho}{M_{\mathrm{H_2O}}} \, \mathrm{d}z \quad . \tag{6}$$





With this, the quantity $X\mathrm{CO_2}$ can be calculated by inserting the Equations 4, 5 and 6 into Equation 1.

As mentioned in Section 2.1, the solar FTIR observations integrate along the actual line of sight towards sun, which generally is not along the vertical. So in fact slant columns are measured and converted to vertical columns. The model column is calculated without considering this complication. Since only enhancements in $X\mathrm{CO_2}$ introduced by local sources and sinks are simulated, all the contributions to $X\mathrm{CO_2}$ reside close to the surface, making the influence of the variable slant line of sight negligible. In addition, near the surface the column sensitivity of the remote sensing observation is close to 1; the median

sensitivity is 1.1 for the lowermost layer. For this reason, the column sensitivity is not taken into account when evaluating the simulated columns.

## 2.3    Harmonized dataset

The simulated and observed datasets need to be harmonized for comparison. For this purpose, $X\mathrm{CO_2}$ values at the observation sites are inferred from the interpolation of the simulated $X\mathrm{CO_2}$ fields. Periods without observation data (e.g. nighttime, or

observation gaps) are removed from the simulated site data.

     The beam of sunlight entering the EM27/SUN covers a cone with a diameter of 0.3 deg. The output of the simulation represents the average over a grid cell. At the spatial resolution of the simulation of 1 km, the derived column is therefore the average over the area of $1\,\mathrm{km^2}$. Therefore, compared to the simulation the observations can be considered as a pencil-beam measurement. Spatial averaging is not possible for the observational data which, by definition, were collected only along

individual pencil lines of sight. To overcome this inconsistency, temporal averaging is applied to the observational data, as the lateral wind transport physically averages the concentrations spatially along the wind direction by mixing and advecting small fluctuations. We have chosen one minute integration time for the EM27/SUN FTIR measurements. In a post-processing step, the observation data was averaged in 10 min bins. For the median wind speed observed at the campus of $1.54\,\frac{\mathrm{m}}{\mathrm{s}}$, the wind traveled 924 m in the period of 10 min, approximately matching the resolution of the simulation grid. The same output

frequency is chosen for the simulation.

     Since only the enhancement from local sources and sinks was included in the simulation, the simulated and measured datasets have different background levels. To estimate the measurement background level of a given day, the 5% quantile $Q(5)$ was calculated and subtracted from the observed data points. This was analogously applied to the simulated time series. Under the assumption that the variations in $X\mathrm{CO_2}$, that are not induced by local sources and sinks, can be considered as constant

over the time span of some hours, the background offset between both datasets is removed by this method. This enhancement over the background achieved by subtracting $Q(5)$ will be called $X\mathrm{CO_2'}$ in the following.

## 2.4    Pixel-wise scaling

To be able to re-scale the inventory after the runtime of the simulation we divided the inventory spatially into different parts. The part of the city with the highest emissions is separated into pixels of $4\,\mathrm{km^2}$, each containing 4 pixels of the simulation grid. This

separation of the city area into pixels is shown in Figure 10A. Each pixel emits into a different tracer (e.g. `CO2_pixel1`), where the simulated $\mathrm{CO_2}$ from the city emissions as stated by ODIAC is then the sum of all these pixel tracers. A altered





prediction can be calculated from the simulation, using a re-scaled emission map. The re-weighting factors $x_i$ can be found by optimizing to best-match the observations by minimizing the cost function

$$c = \frac{1}{2} \cdot || \sum_i x_i \mathbf{a_i} - \mathbf{b} ||^2 \quad , \tag{7}$$

where $\mathbf{b}$ is the observation vector (containing the the observations of both spectrometers) and $\mathbf{a_i}$ is the corresponding vector originating from pixel $i$, interpolated to the sites of the observation. The algorithm used was the bounded variable least square solver (bvls), implemented in the `scipy` library in the function `optimize.lsq_linear` (Virtanen et al., 2020).

Many previous emission inversion studies used the Bayesian inversion framework described by Rodgers (2000) (see e.g. Ohyama et al. (2023); Viatte et al. (2017)). The benefit of the least-squares approach is, that it is easy to implement and does
not rely on an accurate description of the prior's uncertainty, which is not provided for the ODIAC inventory.

We used two different configurations for optimization:

1. *city scaling* – joint scaling of all city pixels with the same factor,

2. *pixel scaling* – individual scaling of each pixel.

Outside the subdivided area the emissions of the ODIAC inventory remain unchanged. The scaling factors, were restricted so
that the prior could only be scaled in the range (1/6, 6), that was empirically chosen. This implies also that a change of sign was not possible in the optimization, and all emissions were required to be positive.

All operations applied to the observed and to the simulated data are shown in a flowchart in Figure 5. This method provides an alternative to calculating trajectories in a Lagrangian model, as for example was done in the data analysis of the Tokyo campaign (Frey et al., 2019). Instead, the results of the forward model ICON-ART can be used directly and without changes
to the model.

## 3 Results and discussion

### 3.1 Calibration and background variability

The two spectrometers were calibrated for the campaign using side-by-side observations. We estimate an systematic calibration uncertainty of 0.22 ppmv (see Appendix A). To understand the impact of spatial heterogeneity and to test if the daily variations
are dominated by emission sources in the close proximity, two of the measurement days were dedicated to measurements with a small distance of approximately 500 m. The variations of these observations are compared to the side-by-side observations. Figure 6 shows the difference plotted against the mean of two corresponding observation points for both, the side-by-side and the small distance observations. The measurement at 500 m distance has a slightly higher standard deviation of 0.19 ppm compared to the side-by-side data (0.15 ppm). Additionally, a small bias of 0.1 ppm is apparent in this observation, which
is below the estimated uncertainty of the calibration. In conclusion, the comparison shows that the 500 m observations are



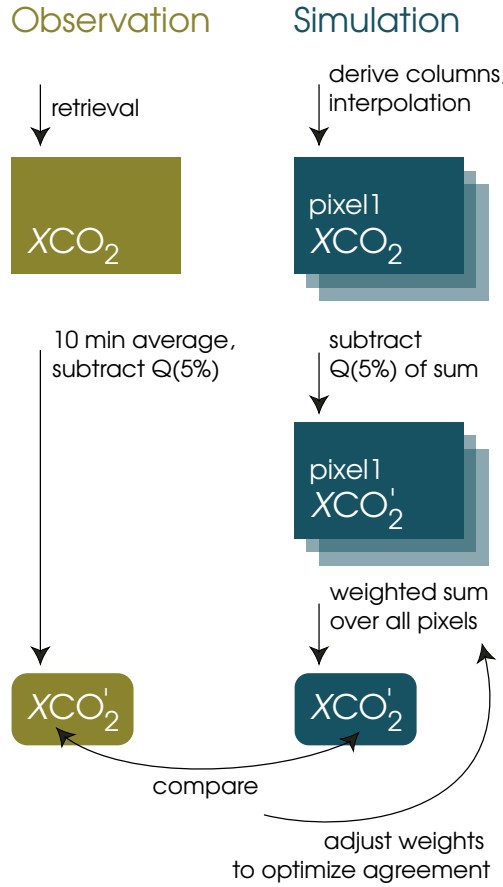

**Figure 5.** The model-observation comparison scheme. The operations applied to the observed data is shown in the green (left), the operations applied to the simulated data are marked in blue (right). The quantity for comparison $XCO_2'$ is deduced from the observed and simulated data. Weights for the individual pixels are derived by re-weighting the pixels of the simulation to best represent the observed $XCO_2'$.

comparable to the side-by-side measurements. This supports the assumption that the observed variations are in fact coming from emissions that can be spatially resolved in the 1 km resolution ODIAC inventory.

By scaling only the city emissions, we assume that the daily variations are dominated by the sources of the city, and not by sources at a greater distance. This assumption was tested with an idealized ICON-ART simulation by investigating in reverse how much the local $XCO_2$ is influenced by the city emissions at four locations at a distance of approximately $70\,\text{km}$ from the city. The enhancements due to the city emissions are compared to the enhancements from the whole ODIAC inventory in Figure 7, showing that the city emissions do not dominate the $XCO_2$ enhancements, even though the city of Thessaloniki is a larger source than the emissions near the rural points that were investigated. We conclude that the emission sources far from the observer play a minor role.



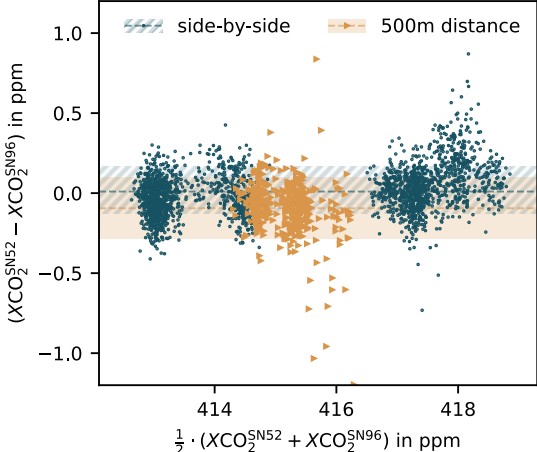

**Figure 6.** Analysis of observed differences between the two spectrometers. All coincident observations points are shown depending on their average on the x-axis and their difference on the y-axis. The side-by-side measurements that were used for calibrating both spectrometers are shown next to measurements recorded to asses the background variability. They latter were recorded with a smaller distance of 500 m to test if a significant part of the variability is induced by sub-gridscale effects. The side-by-side observations are displayed with blue dots and the 500 m distance observations with orange triangles. The average of all points is displayed as dashed line and the standard deviation $\sigma$ is marked through a transparent area ranging from $-\sigma$ to $+\sigma$ starting at the average. The 500 m distance observation shows a small bias of 0.1 ppm. Overall the agreement between both spectrometers for the 500 m distance observation is comparable to the side-by-side observations.

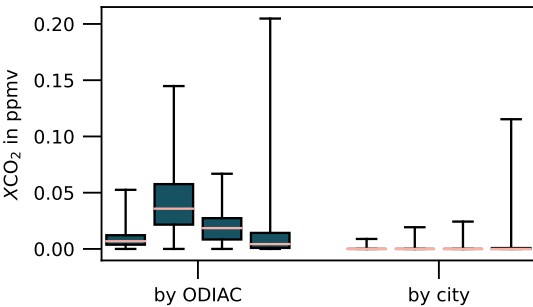

**Figure 7.** Investigation of the influence of distant sources. The modeled enhancement of $X\mathrm{CO_2}$ at four observation points at a distance of approximately 70 km from the city are shown as box plots. The four points are defined by the corners of a rectangle around Thessaloniki with an edge length of 1 °. The corners of the boxes range from the 25 to the 75 percentile, the whiskers indicate the minimum and maximum values. The enhancement from the whole ODIAC inventory (left) is compared to the enhancement that purely originates from the city of Thessaloniki's emissions (right).

### 3.2 Co-observed meteorological variables

To test whether the model simulates realistic meteorological conditions, the model results are compared with the observed wind data and the water vapor column $X\mathrm{H_2O}$, which were co-observed by EM27/SUN. Both quantities are independent of the



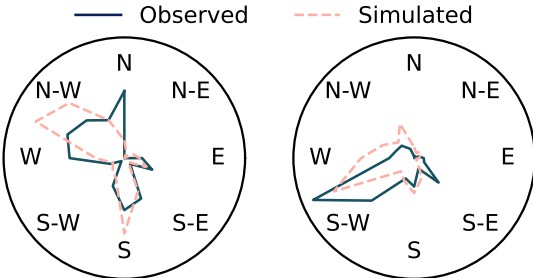

**Figure 8.** Comparison between the observed and simulated wind at two of the observation sites: The Airport (left) and the Thermi (right) (compare Figure 2). The comparison shows the observed and simulated wind roses in the time period between 3 and 18 UTC. The distance between the two sites is only 5 km, which are only a few grid cells in the model.

**Table 2.** Comparison of the observed and simulated wind and water content of the atmosphere. The Pearson correlation coefficient $r$, the mean value of the observation $\bar{x}_{\mathrm{OBS}}$, the standard deviation of the difference between the observed and simulated distributions $\sigma(x_{\mathrm{OBS}} - x_{\mathrm{SIM}})$ and the bias between the simulated and observed datasets $\overline{x_{\mathrm{OBS}} - x_{\mathrm{SIM}}}$ are shown for the available observations. The unit of the three rightmost columns is $\frac{\mathrm{m}}{\mathrm{s}}$ for $u$ and $v$, and ppmv for $X\mathrm{H}_2\mathrm{O}$. The instrument letters for the meteorological observations are listed in Table 1.

| Instrument | Variable | $r$ | $\bar{x}_{\mathrm{OBS}}$ | $\sigma(x_{\mathrm{OBS}} - x_{\mathrm{SIM}})$ | $\overline{x_{\mathrm{OBS}} - x_{\mathrm{SIM}}}$ |
|---|---|---|---|---|---|
| B | u | 0.70 | 1.23 | 1.46 | -0.22 |
| B | v | 0.66 | 0.59 | 1.68 | 0.31 |
| C | u | 0.55 | 0.75 | 1.07 | -0.16 |
| C | v | 0.79 | 0.59 | 1.23 | 0.34 |
| D | u | 0.70 | 0.95 | 1.40 | -0.01 |
| D | v | 0.83 | 0.32 | 1.68 | 0.18 |
| SN52 | XH2O | 0.96 | 3976.73 | 324.19 | -45.28 |
| SN96 | XH2O | 0.96 | 3887.77 | 339.25 | -87.07 |
| SN52-SN96 | XH2O | 0.33 | 52.18 | 166.87 | 45.34 |

carbon dioxide emissions and therefore are suitable to asses the agreement. Especially $X\mathrm{H}_2\mathrm{O}$ is interesting, as it is observed at the same locations as the quantity of interest $X\mathrm{CO}_2$, and the $X\mathrm{H}_2\mathrm{O}$ observations have a high precision and accuracy. Tu et al. (2021) found a dry bias of up to 4.64 % in comparisons against MUSICA IASI satellite observations and a wet bias of up to 9.71 % in comparisons against TROPOMI, while the observed correlations show good agreement between the observations.

First, we compare the observed and simulated wind in the model for validation. The time period between 3:00 and 18:00 UTC for all simulated dates is considered for this comparison. Overall, the observation was well matched by the simulation, which reproduced the complex spatial structure. This can be seen most expressively by comparing the wind directions from Thermi and the Airport that have a distance of 5 km. The wind roses for both sites are shown in Figure 8. Although only a few pixel separate the two observation sites, the wind roses differ significantly, which is well reproduced by the model. The wind



**Table 3.** The resulting total emissions and the improved agreement in the $XCO_2'$ time series are listed for the tested configurations. The first two columns describe the different configurations, using different scaling approaches (see Section 2.4) and sub-samples (see Section 3.4). After applying the rescaling using the different approaches, the total emissions of the city area, the correlation of the resulting time series of $XCO_2'$ ($r$), and the least-squares cost function ($c$) that is defined in Equation 7 are shown.

| | | emissions (Mt/yr) | r | c |
|---|---|---|---|---|
| whole sample | no scaling | 1.9 | 0.10 | 158.9 |
| | city scaling | 2.9 | 0.17 | 144.8 |
| | pixel scaling | 2.9 | 0.29 | 125.4 |
| loose selection | no scaling | 1.9 | 0.36 | 89.9 |
| | city scaling | 3.4 | 0.42 | 68.1 |
| | pixel scaling | 3.0 | 0.53 | 57.3 |
| strict selection | no scaling | 1.9 | 0.34 | 49.1 |
| | city scaling | 4.4 | 0.66 | 24.5 |
| | pixel scaling | 3.7 | 0.77 | 17.1 |

observation at all sites correlate with the simulation. However, the difference between the simulated and observed wind speed show a wide variation and there is a bias. The bias and the standard deviation of the difference are more pronounced when examining the variable $v$, for which consistently lower wind speeds were observed than for the variable $u$. All comparison
quantities for the wind are listed in Table 2.

The simulated water vapor column corresponds very well with the observations, with correlations greater than 96 %. The observed bias is 87 ppmv which is less than 5 % of the average observed $XH_2O$ of 1943 ppmv for the portable spectrometer. While their absolute concentration was very well matched, the correlation decreases to only 33 % when examining the differences between the two observing sites $\Delta XH_2O$. The values for comparison between observed and simulated water columns
are shown in Table 2 as well.

We conclude from the comparison, that the overall meteorological situation is generally well represented, but especially when looking at small scale variations, discrepancies become apparent. We therefore also expect some deviations to appear in the comparison of $XCO_2'$.

### 3.3 Time series of $XCO_2'$

The comparison between the simulated and observed time series of $XCO_2'$ is the basis for investigating the city emissions. The observed time series as well as the simulated time series are shown in Figure 9 in the top row of Panel A and B respectively. When comparing the simulation without scaling the emissions, large discrepancies are visible. The original simulation and the measurements are correlated with a Pearson correlation coefficient of $r = 0.1$, illustrating the large discrepancies.






**Figure 9.** The observed time series of $XCO_2'$ for both spectrometers are shown together with the unscaled simulation (orange dashed line), and both scaling approaches, the pixel scaling (solid blue line) and the city scaling (dash-dotted green line). Panel A shows the observations of the stationary spectrometer SN52, panel B shows the observations of the spectrometer SN96 located at various positions. The top row of both panels shows the whole dataset, the lower rows show the different sub-samples that were investigated (see Section 3.4).





In addition to the simulation without any scaling of the ODIAC inventory, Figure 9 shows the time series after applying the

city scaling and the pixel scaling approaches that are described in Section 2.4. The correlations of the $XCO_2'$ time series for

the re-scaled datasets are significantly improved up to 0.29 (0.17) for the pixel-scaling (city-scaling) approach. While both, the

city scaling and the pixel-wise scaling improve the sample, large discrepancies to the model remain with both scaling methods

applied. The correlations between the simulated and observed $XCO_2'$ time series and the least-squares cost functions (see

Equation 7) are listed the first segment of Table 3.

It is also noteworthy that on some days (e.g. on 2021-10-22) the scaling has a strong effect on $XCO_2'$, visible by the large

discrepancies between the original and the rescaled curves. At the same time, on other days (e.g. on 2022-07-07) the unscaled

and optimized curves differ only slightly. This indicates that the variability in the simulated data is dominated by the biogenic

source and sink for these days. Therefore the optimization has little effect on the resulting $XCO_2'$ time-series as only the

anthropogenic component is altered but the biogenic contribution is not scaled with respect to the atmospheric observation.

Both scaling approaches result in an increase of the anthropogenic emissions. The resulting emissions are discussed in

Section 3.5).

### 3.4 Investigating sub-samples

The correlation between modeled and observed $XCO_2'$ varies to some extend, when looking at individual days. To test the

robustness of the fit results, two sub-samples were considered, selecting days where the correlation of the $XCO_2'$ time series

of the respective day was higher than a given threshold.

1. *loose selection* ($r > 0.2$)

2. *strict selection* ($r > 0.6$)

The loose selection contains 11 days of measurements, while the strict selection criterion was met on only 5 days. The city

emissions were scaled in a fit, as described in the previous section. The correlation, cost function and total emissions after

scaling are listed for the two sub-samples in the second segment of Table 3. Both sub-samples are shown in the two lower

panels of Figure 9A and 9B respectively.

As is to be expected, when comparing all values of the simulated and observed $XCO_2'$ time series in the selected samples,

the two sub-samples have a higher correlation and smaller cost function compared to the whole dataset, when looking at the

unscaled simulation. The very limited correlation of 0.1 improves to 0.34 for the strict selection. For all three samples the

agreement can be significantly improved by both scaling approaches, the highest correlation of 0.77 can be seen when scaling

the individual pixels and optimizing with respect to the strict selection.

In Table 4 also the observed and simulated water vapor columns are compared for the sub-samples. Compared to the whole

sample, the correlation between observed and modeled $XH_2O$ increases, especially when considering the difference between

both spectrometers. The correlation of $\Delta XH_2O$ is only 0.33 for the whole sample and increases to 0.66 for the strict selection.

This is remarkable because, $XH_2O$ is not influenced by the emissions. This is indicating that the uncertainties in the emissions

are not the only relevant source of uncertainty. A possible explanation would be, that the local transport inside the city is often





**Table 4.** Correlation between the simulated and observed $X\mathrm{H_2O}$ time series for the sub-samples. For each sample, the Pearson correlation coefficient is listed for both spectrometers individually and for the difference between the two spectrometers.

| Instrument<br>Sample | SN52 | SN96 | SN52-SN96 |
| --- | --- | --- | --- |
| whole sample | 0.962 | 0.958 | 0.330 |
| loose selection | 0.986 | 0.987 | 0.209 |
| strict selection | 0.995 | 0.994 | 0.664 |

not well represented by the model when looking at such small time-intervals and distances, influencing both the agreement in water vapor and the time dependency of carbon abundances originating from local sources. We discuss the possible sources of the discrepancies in Section 3.6.

**3.5 Re-scaled emissions**

The scaling factors derived by re-scaling the time series of $X\mathrm{CO_2}$ result in a re-scaled emission inventory. The resulting emission maps are shown in Figure 10A. The three plots in the top column show the city scaling approach, where the city was scaled as a whole. For the pixel scaling (the three plots in the lower column of Figure 10A) it is visible that some pixels receive low weights while others receive much higher weights. Comparing the three resulting emission maps resulting from the pixel

scaling, some of the pixels show high emissions for all configurations whereas some pixels are also differing clearly between the configurations. This indicates that not all parts of the found emission map are robust with respect to the chosen configurations. It is still are remarkable fact, that some of the possible hot spot emissions indicated in Figure 10B are consistently receiving high weights. This makes a detection of sources plausible, especially for the following source regions.

– The pixels around the harbor where also the city is most densely populated are receiving high weights in all samples.

– Also the westernmost pixel is consistently receiving high weights. The point is close to the refinery, but it does not match the position exactly. A possible reason might be the wrong height attribution of the tracers as the ODIAC inventory does not provide any height information of the sources. Therefore all emission are simulated at the surface level, while in reality, the emissions from the refinery are emitted from chimneys. The vertical mismatch of the emission height might by compensated in the optimization by a horizontal mismatch.

– The pixels closest to the airport receive high weights, although the number of up-weighted pixels in the south of the city fluctuates. Only the region of the cement factory shows no enhancement in the emission maps presented in Figure 10A.

From this robustness of the fit in different configurations, we conclude that a detection of the most prominent sources from the observation dataset is plausible.

In addition to looking at the distribution of the rescaled pixels, we considered the amount of total emissions, integrated over

the city area. The total emissions for the different optimization configurations is summarized in Figure 10C. The different





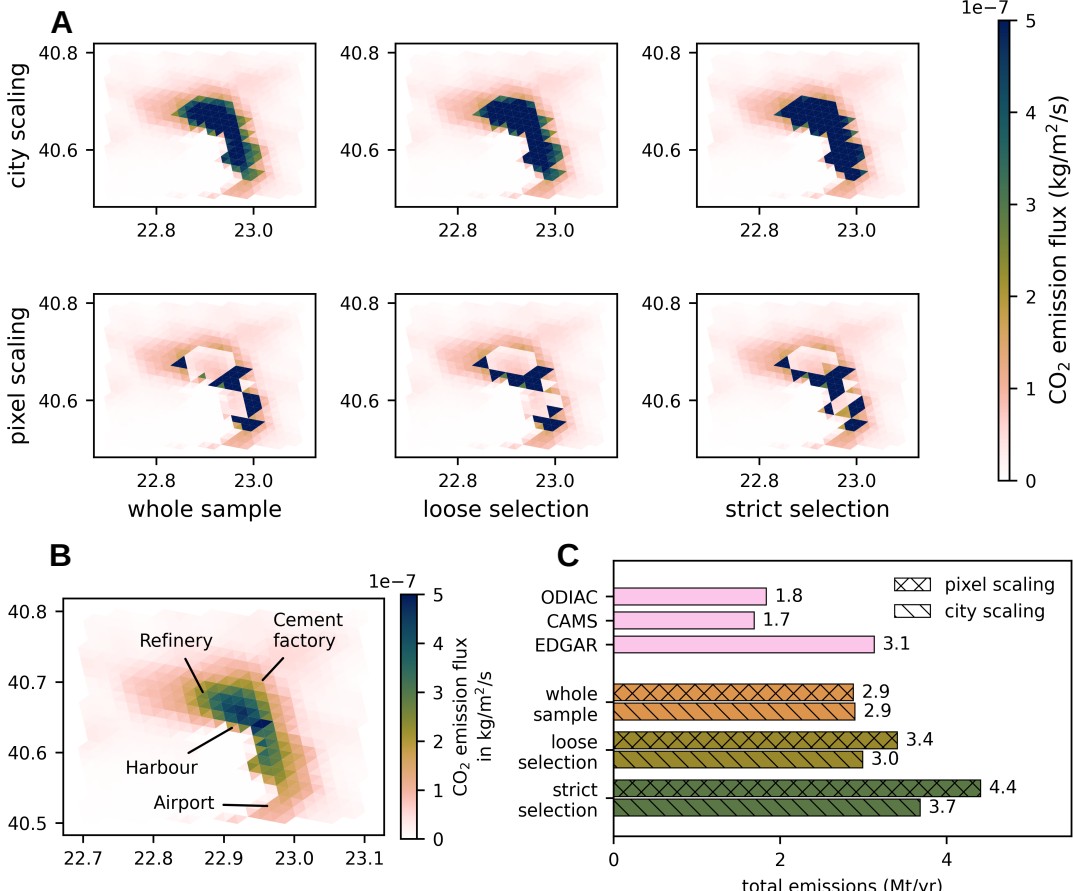

**Figure 10.** Rescaled Emissions. Panel A shows the rescaled emission maps for the different samples and scaling approaches. The three columns correspond to the different samples, and the two rows show the results of the city scaling and the pixel scaling, respectively. The original ODIAC inventory and its separation into scalable pixels are shown in Panel B. Expected hot spots are indicated. Panel C shows the sum of the emissions in the predefined city area for the different scaling approaches and samples as a bar chart, together with the total emissions of the three bottom-up inventories.

optimization configurations predict emissions between 2.9 and 4.4 Mt/yr. While this is a large spread of emission estimates, all configurations are predicting consistently higher emissions. Taking into account that a significant improvement of the agreement between observation and simulation can be seen with this higher emission estimates, the ODIAC inventory is likely to underestimate Thessaloniki's actual emissions for the observation period.

### 3.6 Limitations of the method

Here, we will discuss the possible sources of discrepancies between simulated and observed $XCO_2'$ that go beyond the accuracy of the inventory.



**Model resolution**

The systematic limitations of the model resolution was discussed in Section 3.1. In comparison between site-by-site measure-
ments and observations with a distance of $500\,\mathrm{m}$, we see no significant emission signatures. We conclude that sub-scale effects
do not dominate the observed time series of $X\mathrm{CO}_2'$. Therefore, we consider that resolution does not play a major role.

**Background Estimation**

The method for background estimation using the 5% quantile relies on the assumption that the far-field background does
not vary significantly between the observation sites and that the background is constant over the course of the day. For the
first assumption, the contribution from sources outside of the domain can be assumed to be well mixed at the scale of the
city extend. Also this far-field contribution can be expected to be small, this assumption was also tested in Section 3.1. The
assumption of a constant background over time can not be tested with the available datasets. A trend in the background could
lead discrepancies in $X\mathrm{CO}_2'$ that might look like a signal from local emissions on a given day. However, under the assumption
that the far-field background does not have the same diurnal time dependency, the optimization is not capable of adapting to
these variations, as the emissions can also be scaled for the whole time series simultaneously. We expect that days with a high
influence of a variable background concentration also show higher discrepancies before scaling and are filtered when selecting
the sub-sample.

**Meteorological conditions and transport**

The meteorological discrepancies can be investigated by comparisons to the wind and $X\mathrm{H}_2\mathrm{O}$. In Section 3.2 we argue that
the meteorological situation is basically well matched. Nevertheless, we find that although the observed absolute $X\mathrm{H}_2\mathrm{O}$ is
very well represented by the simulation, the difference between the observation sites is not. This suggests that the transport
is a relevant source of uncertainty. This point is strengthened by the fact that the correlation in $X\mathrm{H}_2\mathrm{O}$ improves in the sub-
samples, that were selected by the correlation between observed and simulated $X\mathrm{H}_2\mathrm{O}$. A possible explanation for this notice
would be that at days with a better agreement between observed and simulated $X\mathrm{CO}_2$, the transport was better matched by the
simulation, which results in a better agreement in $X\mathrm{H}_2\mathrm{O}$ as well.

A second source of uncertainty of the transport comes from the relatively short simulation time. Enhancements from long
transportation times cannot be simulated.

**Sources and sinks**

One of the main sources of discrepancies is likely to come from the emission inventories. The spatial discrepancies between
different anthropogenic inventories are discussed in the Introduction and also temporal discrepancies are possible. For the
anthropogenic part of the emissions, the spatial discrepancies can be reduced by the re-scaling inside the city area. However,
the emissions outside the city area and the temporal variations remain unchanged. A mismatch would probably effect the daily
variations of $X\mathrm{CO}_2'$ significantly.



Likewise, the discrepancies introduced from sources and sinks by the biogenic contribution are not altered by the optimization method. We discuss in Section 3.3 that the discrepancy from the biogenic part of the emission sinks and sources are the most relevant for some of the days. One effect that is for example not included in the model is, that plants in the Mediterranean are able to close their stomata in hot and dry conditions to reduce water loss. This leads to a decrease in gas exchange rates and a depression in the NEE (Lange et al., 1985). This effect is not present in the averaged diurnal cycles from the FLUXCOM X-BASE dataset (compare Figure 4C).

However, the re-scaling of the anthropogenic part of the emissions inside the city area resulted in a significantly improved agreement between the time series of the simulated and observed $XCO_2'$ time series, showing that the city emissions were a relevant contribution and the mismatch could be reduced by the re-scaling method.

## 4    Summary

We present results of a field campaign in Thessaloniki, Greece, performed in October 2021 and summer 2022. During the campaign two solar FTIR spectrometers (EM27/SUN) were operated in parallel from different positions in the city. One of the instruments was permanently positioned at the Campus in the city center, the other instrument was equipped with solar power supply for portability. It was transported to various locations around the city. A time series of the column-averaged molar fraction $XCO_2$ of 179 hours over 22 days was recorded during the period, taking into account both instruments and excluding the measurements taken for calibration. The numerical weather prediction model ICON-ART was used for simulating the enhancements originating from the anthropogenic and biogenic net emissions. The high-resolution ODIAC inventory is used for the anthropogenic emission flux of $CO_2$. The biogenic net emission flux is combined from three different datasets.

As expected, the comparison between the simulated and observed datasets show discrepancies. These could be reduced significantly by re-scaling emissions of the city of Thessaloniki in the ODIAC inventory. Different scaling approaches were compared and the method was additionally tested with two smaller samples to test its robustness. In all of the tested configurations the disrepancies to the atmospheric observations could be improved significantly. However, considerable disrcepancies remain in all configurations. We discuss the possible reasons why these discrepancies remain after optimizing Thessaloniki's anthropogenic sources. We identify the short simulation time, limitations in simulation of the transport, missing time dependency of the anthropogenic emissions and large uncertainties in the estimate for the net ecosystem exchange as relevant factors. Especially the representation of the biogenic interactions seem to dominate the agreement in some periods of the observation. Further studying this sources of limitations could significantly improve the ability to accurately reproduce the atmospheric observations.

In the pixel scaling approach, where different weights were assigned for different parts of the city, the area around the harbor, a pixel close to the refinery and the part closest to the airport consistently receive high weights in the different samples. This indicates that a plausible detection of sources is possible with our method.

The pixel scaling and also the city scaling, where all pixels were required to have the same weighting factor, result consistently in an increased estimate for the total emissions of Thessaloniki, for all tested configurations. The predicted emissions





after re-weighting range from 2.9 to 4.4 Mt/yr for the city area, which is significantly higher than the emissions stated by the ODIAC inventory (1.8 Mt/yr). This indicates that the original ODIAC inventory underestimates the emission during the period of observations.

*Data availability.*  The collected observational dataset of the COCCON Campaign in Thessaloniki is accessible at https://doi.org/10.48477/COCCON.PF23.THESSALONIKI-CAMPAIGN.R01. The corresponding simulation dataset that was produced for this work can be accessed at https://doi.org/10.5281/zenodo.12666197. Secondary data sources are cited in the text.

**Appendix A: Calibration**

Side-by-side measurements were performed at the beginning, middle and end of the campaign period for calibrating the spec-
trometers. In total 5 days of calibration measurements were collected. Side-by-side measurements at Karlsruhe with the COC-CON reference spectrometer were used to calibrate both spectrometers to the Karlsruhe TCCON station (compare Alberti et al. (2022)).

From these measurements, the empirical correction factors $K$ were calculated for both campaign spectrometers, where the corrected $X\mathrm{CO}_2$ is calculated by

$$X\mathrm{CO}_2 = X\mathrm{CO}_2{}^{\mathrm{raw}} \cdot K_{\mathrm{CO}_2} \quad .$$  (A1)

For the derivation of the correction factors, the observation times are harmonized by constructing a time series with averages over 5 minutes. The correction factors for the spectrometers A and B are calculated so that

$$1 = \mathrm{mean}\left(\frac{X\mathrm{CO}_2{}^{\mathrm{A}}}{X\mathrm{CO}_2{}^{\mathrm{B}}}\right)$$  (A2)

is fulfilled when looking at the whole sample of calibration observations.

To estimate the systematic uncertainty due to the calibration, a relative calibration factor between both spectrometers was derived for each day of the side-by-side measurements in Thessaloniki individually. The difference between the maximum and minimum relative calibration factor is 0.05 % which corresponds to an uncertainty of 0.22 ppmv for the calibration period.

*Author contributions.*  LF prepared the manuscript, initialized the field campaign, set-up the simulation and conducted the analysis. LF, PS, MM and DB and FH discussed the detailed implementation of the field campaign and performed the EM27/SUN observations. JG developed, 420  prepared and tested the solar power supply for the second spectrometer. BH prepared the Vaisala pressure sensor for pressure recording in Thessaloniki. PS constructed the reference pressure record and derived the pressure offsets between the observation sites. CA provided the instrumental parameters. LF and DD preformed the retrieval of the raw observation data. LF, SV and RR developed the simulation setup. RR, PB, FH and LF developed the idea to pixel-wise re-scale the original emission inventory, and discussed the implications of the discrepancies between simulated and observed wind, $X\mathrm{H}_2\mathrm{O}$ and $X\mathrm{CO}_2'$. All authors contributed to the final editing of the paper.



*Competing interests.* The authors declare that they have no conflict of interest.

*Acknowledgements.* This research was funded by the Helmholtz European Partnership for Technological Advancement (HEPTA) project [Grant Agreement No. PIE-0016]. The Graduate School of Climate and Environment (GRACE) financially supported the travel to Thessaloniki. The simulations for this work were performed on the HoreKa supercomputer funded by the Ministry of Science, Research and the Arts Baden-Württemberg and by the Federal Ministry of Education and Research. This work was performed with the help of the Large Scale

Data Facility at the Karlsruhe Institute of Technology funded by the Ministry of Science, Research and the Arts Baden-Württemberg and by the Federal Ministry of Education and Research.



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
