# Peer review of "XCO2 observations compared to km-scale ICON-ART simulations indicate an underestimation of Thessaloniki's emissions in the ODIAC inventory"

_EGUsphere, 2025_

## Referee Comment (RC2)

The manuscript by Lena Feld et al. presents results from a field campaign conducted in Thessaloniki, Greece, aimed at estimating $CO_2$ emissions using two ground-based remote sensing FTIR spectrometers (EM27/SUN). Simulated $XCO_2$ was generated using the ICON-ART numerical weather prediction model, incorporating anthropogenic emissions from the high-resolution ODIAC inventory and biogenic fluxes from a combination of three datasets. $CO_2$ emissions were then inferred by scaling the modeled $XCO_2$ to match the observations.

This study demonstrates a valuable application of the ICON-ART model supported by ground-based total column measurements. The use of such measurements helps to reduce uncertainties in current emission estimates and shows potential for extension to other trace gases. However, some aspects, such as the uncertainty and sensitivity of the model need further investigation. This study can be considered for publication after the authors carefully address the concerns outlined below.

**Specific comments:**

Line4&5: Please spell out the full names of 'EDGAR' and 'ODIAC' upon first mention in the abstract for clarity, especially for readers who may not be familiar with these acronyms.

Line 118: The reported $XCO_2$ enhancements reached up to 2.03 ppm. I agree with the other reviewer that this signal appears relatively modest. Could the authors clarify the distance between the two instruments, as well as the wind conditions on that particular day? It is possible that the relatively small enhancement was influenced by the instrument spacing, mild wind speeds, or variable wind directions.

It would be helpful to compare this result with findings from similar urban campaigns. For example, in the early Berlin study (doi:10.5194/amt-8-3059-2015), a moderate $XCO_2$ enhancement of approximately 4 ppm was observed at the downwind site.

Line 124: The authors mentioned that simple models, such as the box model, are not suitable for this study and instead proposed a more precise simulation using ICON-ART. Could the authors clarify whether ICON-ART has previously been applied to greenhouse gas or trace gas emission studies? If so, referencing relevant prior work would help support its use in this context.

Line 141: the authors state that they "set the surrounding background concentration to 0 ppmv". Could the authors provide more justification for this assumption? What are the characteristics of the surrounding areas, and is there evidence to suggest they have no significant influence on the target region? Including a spatial distribution of $XCO_2$ derived from satellite observations (or from CAMS $XCO_2$ predictions?) could help assess whether notable emission sources are present in the surrounding areas and validate this assumption.

Line 199: The purpose of the spatial regridding is not entirely clear for me. Specifically, it's unclear why four 1 km × 1 km pixels (from the original ODIAC inventory, which already matches the simulation resolution) are merged into larger 4 km² pixels for the city with the highest emissions. Since the ODIAC inventory has a comparable resolution to the model grid, why not retain the original 1 km × 1 km pixel resolution for emission allocation? Please clarify the rationale behind this regridding step and explain how it supports the goal of rescaling the inventory after the simulation runtime.

Line 214: The sentences of "Outside the subdivided area the emissions of the ODIAC inventory remain unchanged. The scaling factors, were restricted so that the prior could only be scaled in the range (1/6, 6), that was empirically chosen." are unclear to me. Could the authors clarify why emissions outside the subdivided area were kept constant? Additionally, why restricting the scaling factors to the (1/6, 6) range, and how was this empirical range determined?

Line 218-219: I do not think the cited paper by Frey et al., 2019 is relevant to the discussion of trajectories or the Tokyo campaign.

Line 224: the authors use two days of closely spaced measurements (~500 m apart) to assess the impact of spatial heterogeneity. Based on Figure 2, I assume the first instrument was located at Campus and the second at Seych Sou, which appears to be a more mountainous region. It would be helpful to explicitly state the locations of these measurements in the manuscript to provide better context for the reader.

Additionally, I'm curious about the wind conditions during these two days. Was there any prevailing wind direction that could have transported emissions from one site to the other? If so, this might have influenced the observed enhancements and should be discussed as a potential factor in the interpretation. A discussion of wind-related differences between the sites could further strengthen the assessment of spatial heterogeneity and its impact on the measurements.

Furthermore, during the calibration days, there appear to be significant biases of up to 1 ppm. Any reasons for this (e.g., could they be related to higher solar zenith angles which should be filtered?). This level of bias is comparable to the ΔXCO2 signal in this study used to estimate emissions.

Line 277-279: "This indicates that the variability in the simulated data is dominated by the biogenic source and sink for these days……" This statement would benefit from additional supporting evidence. Are there any specific indicators, such as back trajectories?

On several days (as 2022-07-07), the simulation aligns well with the observations. Does this suggest that ICON-ART can reliably model biogenic contributions and no significant anthropogenic emissions were generated on these days? Additionally, is it possible to isolate and

evaluate the biogenic and anthropogenic components separately within the simulation to better support this interpretation?

The no scaling XCO$_2$' values appear to be significantly lower in October compared to June and July, which does not align with the observations. Could this higher discrepancy in October relate to an underestimation of biogenic sources in the ICON-ART model? Or could this discrepancy be related to the background removal method? Specifically, the use of a constant daily background based on the 5th percentile may suppress meaningful variability and potentially remove part of the signal of interest. Given that two EM27/SUN instruments were deployed, was it considered to use the upwind site as a dynamic background reference? Alternatively, if satellite data are available, would using upwind satellite observations provide a more representative background than the current approach?

**Technical comments:**

Figure4: please specify the legend for the two different lines in the figure.

Line 135: 0,808 km >>> dot

Line 215: "This implies also that a change of sign was not possible in the optimization, and all emissions were required to be positive." >>> "This also implies that the optimization did not allow sign changes…"

Line 248: observation was >>> observations were.

Line 249: "expressively" >>> This can be clearly seen by…?

Line 252: observation >>> observations, difference >>> differences

---

## Author Comment (AC1)

We thank Reviewer 1 for their and very constructive suggestions for improvement, that helped us to extend the manuscript and improve the simulation results significantly. We will first give a short general summary of the changes. In the following, we will answer the points brought up by Reviewer 1 in detail.

**Summary of the Changes**

We decided to alter the simulation setup and extend the ICON-ART simulations. This decision was based on the question of the validity of the background assumptions. This aspect was brought up by both reviewers. Especially the question if a larger domain size and longer transportation time are necessary could not be answered sufficiently in the framework of the previous simulation setup. The new setup includes a second outer domain, covering a larger area. A longer transportation time was simulated. We describe the new setup in the revised manuscript.

The new setup increased the general agreement between the prior simulation results and the observed XCO2'.

At the same time, the estimated emissions of Thessaloniki slightly decreased.

The statement of an underestimation of ODIAC's emissions is still valid, especially when considering actual ODIAC data for 2021 and 2022, which became available in the meantime in ODIACv2024 dataset.

To answer the question of the effect of out-of-domain sources the former reverse approach to investigate far-field emissions was replaced by separating the far-field background and longer transportation times and by redoing the emission estimation in different configurations. Both effects affected the resulting emission estimates only minimally.

As was suggested by Reviewer 1, we further investigated the biogenic contribution of the variability, which showed that the biogenic sink plays an important role during the summer month in 2022 but is of minor importance for October 2021. This brought us to the idea of separating the whole observation period in monthly sub-periods instead of selecting days with a better prior agreement.

This separation showed that the agreement was superior for October 2021. Concurrently the largest underestimation of ODIAC's emissions could be seen in this month. The underestimation was less pronounced for the summer.

In summary these changes significantly increased the agreement between observed and simulated XCO2. The extended simulation shows that a larger domain or an increase in simulation times have little impact on the resulting emission estimates. Finally, the results could be refined temporarily indicating a stronger influence of the biogenic sink for the summer month. The underestimation of the emissions was found to be most pronounced in October 2021, where also the best agreement between simulation and observation is visible.

**Specific Replies (Reviewer Comments in blue)**

**Main issue:**

The study makes the strong assumption that the background does not change over a given day. All variability during the day is thus assumed to be due to emissions and biospheric fluxes within the model domain. For each day, a background is subtracted from the observations and only the enhancements above this background are compared with the model. Correspondingly, the model only simulates the signals from CO2 fluxes within the model domain, but no background. Treating the model in this way makes sense under the assumption of a daily constant background, but whether this assumption is justified in the first place is highly questionable. The publication provides no convincing evidence that this assumption holds. If this was not the case, any variability in the background would be translated into emission signals, which could substantially affect the

results. Unfortunately, there were no measurements upstream of the city to test the hypothesis. I fully acknowledge that measuring upstream is complicated in a coastal city, but nevertheless it seems to me that it would have been possible to place the mobile instruments outside in one of the agricultural areas to the south or across the bay to the southwest of the city. It is unfortunate that the 2nd instrument was always placed inside the city.

(1) We agree that these assumption are a crucial part and can potentially influence the resulting emissions estimates significantly. To better justify these assumptions and also the chosen setup (the issue of a short transportation time is raised below) we decided to adapt the simulation setup. We think that with the new simulations we can show that the domain size and simulation time chosen in the updated setup are sufficient.

We were restricted in the positioning of the spectrometers, we added more explanation about the site selection.

There are several factors that may lead to a changing background: Advection of air masses differently influenced by vegetation (note that most of the measurements were taken during thegrowing season) or anthropogenic emissions, changes in air mass origin during the day, or changing influences of the land-sea breeze evolving over the day.

(2) As Reviewer 1 also suggested further down, we added a discussion about the different simulated components and the influence of the biogenic fluxes. This revealed that the biogenic flux is in fact a relevant driver of the diurnal variability in the summer month, while its effect in October 2021 was negligible. This investigation brought us to the idea to perform the optimization also using the individual months. We found that the agreement and success of the optimization were much more pronounced in October 2021.

The authors acknowledge the influence of land-sea breezes and the complex topography, but there is no deeper analysis of their potential impact on XCO2. The wind rose at Thermi (Fig. 8) suggests that there was indeed a prominent land-sea breeze during most of the measurement days, potentially amplified by the hills to the northeast of the city. At low altitude, the sea breeze transports air from sea to land, but at higher altitude potentially from land to sea. There could thus be a recirculation of CO2 from Thessaloniki at higher levels. Was the model domain large enough to capture such effects? The full model domain had a width of 2° east-west, but all figures only show a subdomain. It would have been helpful to see a figure presenting the full model domain and its topography to better judge potential influences of land-sea contrasts and topography.

(3) We added a figure of the original domain (Domain 1) and the now extended domain (Domain 2). To illustrate the complexity of the wind situation we also exemplarily show wind fields and the resulting XCO2 plume for one day in Appendix B.

On many days, there was a marked decrease of XCO2 over the course of the day (see Fig. 9). There is no discussion of this important phenomenon at all. Was this due to an increase in wind speeds or due to the take-up of CO2 by the vegetation over the day?

(4) We added a discussion about the influence of the biogenic fluxes and attribute this decrease over the day to these in the summer month. A decrease in the anthropogenic components is not clearly detectable for the summer month.

In my view, a more careful analysis of the mesoscale flow and its impact on CO2 is needed. The analysis presented in Section 3.2 is extremely limited and doesn't provide any insights into the overall flow situation. Furthermore, a more thorough assessment of the hypothesis of a daily constant background is needed, for example by looking at CAMS CO2 global reanalysis data

upstream of the city. I am also missing an analysis of the contribution of biospheric CO2 to the signals shown in Fig. 9. Wasn't this simulated as a separate tracer? Interpreting the differences between model and observations only in terms of emissions from the city but ignoring the influence of potentially wrong assumptions regarding background and biospheric fluxes is very risky, in my view too risky.

(5) To address this, we adapted the simulation setup and tested different domain sizes and transportation times (see also Answer 1).

We attribute the robustness of the fit also to the fact that the fit is performed using the full time series. Outside-domain background might be present but does not have the same structure of the signals induced by the city emissions. Hence, the optimization cannot wrongly interpret this background-induced discrepancies as city signal.

Another issue is that there is no discussion of the strategy of how the station pairs were selected on the different days. Typically, such studies use an upwind-downwind configuration (or at least a positioning along the main wind direction) to quantify the emissions based on the difference in XCO2 levels between the sites. However, it seems that in this study the selection of the mobile site was not guided by actual wind conditions but rather by the idea of covering all parts of the city over

the course of the campaign. This approach needs to be better motivated. In fact, the study makes very little use of the fact that there was a pair of sites. Rather each site contributes to the results independently. The chosen approach would thus be equally valid (or invalid) when applied to a single site, e.g. the stationary site that has been measuring over a much longer period.

(6) We added explanation about the strategy how the observation sites were determined. The data of the two sites constrain the fit results, even if the gradient between the sites is not explicitly fitted. The implicit fit of the gradient by simultaneously optimizing the data can be seen from the fact that the correlation of the modeled and observed gradient improves by the optimization even though it was not explicitly targeted. To show this we added a corresponding column to Table 3.

Another inversion study for a coastal city using total column observations (but not cited) was performed in Los Angeles by Hedelius et al. (https://doi.org/10.5194/acp-18-16271-2018). In their study, they used a TCCON site outside Los Angeles as background and analyzed only the difference to the TCCON site within the city. This seems to me to be a much more robust approach.

(7) We agree that the study performed by Hedelius et al. 2018 has a more robust approach for identifying the LA emissions from a multiple-year time series using a background station. From the TCCON dataset a robust estimate of 104 Tg CO2 yr-1 is derived with a relatively small uncertainty of  $\pm$  26 TgCO2/yr.

This study in Thessaloniki is based on a much smaller dataset, that covers only a shorter period. We were not able to implement a dedicated background station, for different reasons (we discuss constraints in the positioning and the complexity of the wind field). Also the emissions of the city of Thessaloniki are with 3.0 Tg CO2/yr (EDGAR) much smaller than the emission of LA. However, we consider the proposed approach a valuable alternative, to check the consistency of BU emissions estimates. There is a very limited amount of cities whose emissions can be determined from a pair of permanent TCCON stations. The proposed approach by looking not at the long-term statistical trends but the variability of a limited number of individual days, would open up the possibility to monitor the emissions of many more cities with limited instrumentation, and also would allow to spot-check specific cities. Also the consistency of much smaller sources is of interest. We argue that our study - even if no robust estimate but a mere indication of the consistency of different inventories – indicates that the campaign approach is a useful method. Such

indication to find large discrepancies in the BU emission inventories on smaller scales could be adapted to spot-checks of specific sources in the following.

We added a reference to the LA study and a comparison between the different signal strength to the Introduction and also extended the abstract slightly, to underline this difference in size and amount of data available.

**Further Important Points**

- 1. Several data sets used in the study are not described in sufficient detail. These include the CEDS data set mentioned on page 2 and the MODIS, SMAP and FLUXCOM-X-BASE data set mentioned on pages 7/8. Without any further details, their role and utility for the study is difficult to judge.
- (8) We added additional information about the mentioned dataset.
- 2. CAMS has produced the CAMS-REG emission inventory (Kuenen et al., 2024; https://doi.org/10.5194/essd-14-491-2022) that has higher resolution than the CAMS-inventory used in this study. It would make much more sense to use that inventory.
- (9) As we used the CAMS and EDGAR inventories only for comparison and decided for the ODIAC inventory as data source for the prior emissions in the simulation, we did not change the CAMS inventory, to CAMS-REG, which is only available on request. However we updated the used inventories, where a new version became available.
- 3. Figure 3 summarizes the measurements at the two sites. Considering the importance of the gradients between the site pairs, it would be useful to add another panel, in which the median of the instrument at Campus is subtracted from each pair. All box plots for the Campus instrument would thus have a median of 0. Furthermore, I was strongly confused by the labels on top of this figure. As described in the text, one of the instruments remained at the same location (at Campus), but the figure shows two different labels for this instrument, "P" and "M". I also found it confusing that the same site was named differently in different contexts. The campus site, for example, is sometimes called Campus, sometimes P (Physics), and sometimes "A". Why not use the same name throughout? I don't see the point of using a label "A", for example. Writing Campus instead of "A" would not make the publication much longer.
- (10) We extended Figure 3 with a panel where not the median, but the daily 5-percentile of each data set was subtracted from the observations. This way not only the gradients become apparent, but also the diurnal variability; in addition, the background subtraction method is illustrated. We also replaced the abbreviations of the sites by the full site names for clarity. To clarify the two different sites on campus, we also extended Figure 2 by a close-up view on the area of the campus. We also updated Table 1 and removed the instrument abbreviations.
- 4. The tables need to be improved: Table 1 has four columns but only two titles. "Vaisala" or "Davis" seem to be instruments rather than sites. The site name should come first, then the instrument. Why is there no entry for the accuracy of the pressure measurement at C and D? It would also be good to add a column "wind" and show which sites have wind measurement (e.g. with yes/no or an "X").
- (11) We updated the table accordingly, and also changed the content slightly, to avoid confusion: Instead of the available data, we now show the data that was actually used. For example, the airport WMO station also pressure is available, but we only had access to hourly data, which was not

sufficient for the purpose of the retrieval. Further detail is provided in the "Additional data sources" section.

Table 2: There should be separate columns for site and instruments. It doesn't make sense to have a label "B" referring to a site location in a column called "instrument". It would also be good to add a column for the units. When comparing model results with observations, it is more common practice to express the bias in terms of "SIM – OBS" rather than "OBS – SIM".

(12) We changed the first column to a joined Site/ Instrument column, as for the XCO2 measurements the sites vary and adapted the table as suggested.

*Table 4: The word "Instrument" should be centrally placed above the instrument columns (SN52 etc.)*, not to the left.

(13) We changed the alignment of the table so that the Site/ Instrument, the Variable and the Unit columns are centered.

The simulation setup looks far from ideal to me: Running a simulation for each measurement day separately is a valid option, but starting these simulations only at 3 UTC, i.e. shortly before sunrise, doesn't seem appropriate to me. The XCO2 columns shown in Fig. 9 often show a marked decrease over the course of the day starting from a high value in the morning. These high values could be due to CO2 buildup over the night (e.g. due to vegetation respiration fluxes in the domain), which the model is not able to capture when initialized only at 3 UTC. Probably each simulation would have to be run over 2 days with only the 2nd day being used for the analysis.

- (14) We adapted the simulation setup as was described above. The longer simulation time (starting 3 UTC on the previous day) had an impact on both the agreement and the simulation results. Parts of it might also be due to a newer version of ICON-ART, that was used. We could show that a further increase in domain size and starting time did not impact the emission estimates. We thank Reviewer 1 to this very constructive feedback that resulted in a visible improvement of the results.
- 6. The reverse analysis of the potential influence of distant sources on page 11 is awkward. The influence of sources outside the city should be directly available from the simulation, unless no anthropogenic sources outside the Thessaloniki were included. Furthermore, the short simulation time with initialization at 3 UTC may not be sufficient to see impacts at 70 km distance. Finally, the flow is quite strongly directed due to the land-sea breeze and the mountain circulations. A reverse view in such a situation makes little sense.
- (15) We replaced this reverse analysis of distant sources by testing the different domain sizes, which supports the assumption that the more distant background sources do not have a relevant effect.
- 7. The motivation for the measurements at a small distance of 500 m is not very clear. Of course, with increasing distance it can be expected that difference in XCO2 increase. Whatis the conclusion of the fact that these measurements show a slightly larger scatter than the side-by-side measurements?
- (16) As the statistics is similar for both, the calibration and the 500 m distance observations, and we only have taken a limited amount of data in this configuration. We think that it cannot be clearly stated if this scattering is already a weak signal of emissions, or if the scattering was due to

instrumental effects. Also the calibration series shows scattering around 418 ppm (see Figure 6 in Preprint). Note that when applying the binning to 10min bins these reduce to only a few outliers.

We carefully addressed all small points and corrections raised by the reviewer.

---

## Author Comment (AC2)

We thank Reviewer 2 for the very constructive suggestions for improvement, that helped us extending the manuscript and improving the simulation results significantly. We will first give a short general summary of the changes. In the following, we will answer the points brought up by Reviewer 2 in detail.

**Summary of the Changes**

We decided to alter the simulation setup and extend the ICON-ART simulations. This decision was based on the question of the validity of the background assumptions. This aspect was brought up by both reviewers. Especially the question if a larger domain size and longer transportation time are necessary could not be answered sufficiently in the framework of the previous simulation setup. The new setup includes a second outer domain, covering a larger area. A longer transportation time was simulated. We describe the new setup in the revised manuscript.

The new setup increased the general agreement between the prior simulation results and the observed XCO2'.

At the same time, the estimated emissions of Thessaloniki slightly decreased.

The statement of an underestimation of ODIAC's emissions is still valid, especially when considering actual ODIAC data for 2021 and 2022, which became available in the meantime in ODIACv2024 dataset.

To answer the question of the effect of out-of-domain sources the former reverse approach to investigate far-field emissions was replaced by separating the far-field background and longer transportation times and by redoing the emission estimation in different configurations. Both effects affected the resulting emission estimates only minimally.

As was suggested by Reviewer 1, we further investigated the biogenic contribution of the variability, which showed that the biogenic sink plays an important role during the summer month in 2022 but is of minor importance for October 2021. This brought us to the idea of separating the whole observation period in monthly sub-periods instead of selecting days with a better prior agreement.

This separation showed that the agreement was superior for October 2021. Concurrently the largest underestimation of ODIAC's emissions could be seen in this month. The underestimation was less pronounced for the summer.

In summary these changes significantly increased the agreement between observed and simulated XCO2. The extended simulation shows that a larger domain or an increase in simulation times have little impact on the resulting emission estimates. Finally, the results could be refined temporarily indicating a stronger influence of the biogenic sink for the summer month. The underestimation of the emissions was found to be most pronounced in October 2021, where also the best agreement between simulation and observation is visible.

**Specific Replies (Reviewer Comments in blue)**

Line4&5: Please spell out the full names of 'EDGAR' and 'ODIAC' upon first mention in the abstract for clarity, especially for readers who may not be familiar with these acronyms.

We added the acronyms in the text.

Line 118: The reported XCO2 enhancements reached up to 2.03 ppm. I agree with the other reviewer that this signal appears relatively modest. Could the authors clarify the distance between the two instruments, as well as the wind conditions on that particular day? It is possible that the relatively small enhancement was influenced by the instrument spacing, mild wind speeds, or variable wind directions.

We added the wind conditions for the given day, which showed lower-than-usual wind speeds. We expect the signal to be smaller than in previous campaigns focusing on very big cities, as Thessaloniki is reported to have significantly lower emissions. Thereby, Thessaloniki is a more difficult target for the quantification of its emissions. We added a comparison in emission strengths to a previous study targeting LA to the introduction.

It would be helpful to compare this result with findings from similar urban campaigns. For example, in the early Berlin study (doi:10.5194/amt-8-3059-2015), a moderate XCO2 enhancement of approximately 4 ppm was observed at the downwind site.

The gradient is compared to observed gradients from the Tokyo Campaign where gradients up to 9.5 ppm were observed.

Line 124: The authors mentioned that simple models, such as the box model, are not suitable for this study and instead proposed a more precise simulation using ICON-ART. Could the authors clarify whether ICON-ART has previously been applied to greenhouse gas or trace gas emission studies? If so, referencing relevant prior work would help support its use in this context.

We added two references where ICON-ART is used for inversion applications.

Line 141: the authors state that they "set the surrounding background concentration to 0 ppmv". Could the authors provide more justification for this assumption? What are the characteristics of the surrounding areas, and is there evidence to suggest they have no significant influence on the target region? Including a spatial distribution of  $XCO_2$  derived from satellite observations (or from CAMS  $XCO_2$  predictions?) could help assess whether notable emission sources are present in the surrounding areas and validate this assumption.

For a better justification of the assumptions concerning the advected background, we extended the simulation by a broader domain where more emission sources contributed to the background that is present in the simulation. We could show that the emission estimates did not differ significantly between explicitly taking this far-field background into account or considering only the emissions of Domain 1.

Line 199: The purpose of the spatial regridding is not entirely clear for me. Specifically, it's unclear why four 1 km  $\times$  1 km pixels (from the original ODIAC inventory, which already matches the simulation resolution) are merged into larger 4 km² pixels for the city with the highest emissions. Since the ODIAC inventory has a comparable resolution to the model grid, why not retain the original 1 km  $\times$  1 km pixel resolution for emission allocation? Please clarify the rationale behind this regridding step and explain how it supports the goal of rescaling the inventory after the simulation runtime.

The ICON grid is an icosahedral grid, while the other data sources are provided in latitude-longitude gridding. Therefore, a regridding is necessary. To make this more clear, we moved the image of the ODIAC inventory regridded to the Icosahedral ICON grid as used in the simulation from Figure 10 to an earlier position.

Line 214: The sentences of "Outside the subdivided area the emissions of the ODIAC inventory remain unchanged. The scaling factors, were restricted so that the prior could only be scaled in the range (1/6, 6), that was empirically chosen." are unclear to me. Could the authors clarify why emissions outside the subdivided area were kept constant? Additionally, why restricting the scaling factors to the (1/6, 6) range, and how was this empirical range determined?

To clarify the chosen approach, we extended the paragraph.

Line 224: the authors use two days of closely spaced measurements (~500 m apart) to assess the impact of spatial heterogeneity. Based on Figure 2, I assume the first instrument was located at Campus and the second at Seych Sou, which appears to be a more mountainous region. It would be helpful to explicitly state the locations of these measurements in the manuscript to provide better context for the reader.

Both sites were located on campus. To improve clarity, we extended Figure 2 to include a detailed view of the campus.

Additionally, I'm curious about the wind conditions during these two days. Was there any prevailing wind direction that could have transported emissions from one site to the other? If so, this might have influenced the observed enhancements and should be discussed as a potential factor in the interpretation. A discussion of wind-related differences between the sites could further strengthen the assessment of spatial heterogeneity and its impact on the measurements.

The wind conditions for the days were added in the text. Also the complexity of the wind fields was illustrated exemplarily in Appendix B.

Furthermore, during the calibration days, there appear to be significant biases of up to 1 ppm. Any reasons for this (e.g., could they be related to higher solar zenith angles which should be filtered?). This level of bias is comparable to the  $\Delta XCO2$  signal in this study used to estimate emissions.

The  $\Delta XCO2$  shows high discrepancies for individual pixels for both configurations, the 500 m distance measurements, and the side-by-side observations.

As proposed by Reviewer 2, we checked the dependency of the solar zenith angle (see Figure A).

Figure A: Comparison of the SZA dependence of the sideby-side and 500m-distance observations.

For the 500m-distance observations, we find that the highest discrepancies are visible for the highest observed SZAs. however, the absolute SZA for these instances are still lower than 40 degree. As there are not many observations, it could also be that, e.g. the sun was not perfectly centered for a period at the start/ end of the measurement period where the SZA was minimal on that given date. The outliers are similarly pronounced for the side-by-side observations, e.g. in Figure 6 of the Preprint around 418 ppm. The impact of the outliers is much smaller when binning into 10-minutely bins.

The no scaling XCO2' values appear to be significantly lower in October compared to June and July, which does not align with the observations. Could this higher discrepancy in October relate to an underestimation of biogenic sources in the ICON-ART model? Or could this discrepancy be related to the background removal method? Specifically, the use of a constant daily background based on the 5th percentile may suppress meaningful variability and potentially remove part of the signal of interest. Given that two EM27/SUN instruments were deployed, was it considered to use the upwind site as a dynamic background reference? Alternatively, if satellite data are available, would using upwind satellite observations provide a more representative background than the current approach?

The scaling factors are applied to the full time-series, so they are the same for October and June, however with the overall scaling factor the simulated enhancement based on the inventory is visibly lower in October than the observations. We therefore replaced the sub-sample investigation by individually optimizing the months, which indicates a strong annual variability of the discrepancies between inventory and measurements.

We implemented the technical comments raised by the reviewer.